# Longer Time-Scale Variability of Atmospheric Vertical Motion over the Tibetan Plateau and North Pacific and the Climate in East Asia

**Rongxiang Tian** [1,*], **Yaoming Ma** [2,3,4], **Weiqiang Ma** [2,3,4] , **Xiuyi Zhao** [1] **and Duo Zha** [1]

[1] School of Earth Sciences, Zhejiang University, Hangzhou 310027, China; 21838005@zju.edu.cn (X.Z.); zhaduo0000@163.com (D.Z.)

[2] Institute of Tibetan Plateau Research, Chinese Academy of Sciences, Beijing 100101, China; ymma@itpcas.ac.cn (Y.M.); wqma@itpcas.ac.cn (W.M.)

[3] CAS Center for Excellence in Tibetan Plateau Earth Sciences, Chinese Academy of Sciences, Beijing 100101, China

[4] University of Chinese Academy of Sciences, Beijing 100049, China

\* Correspondence: trx@zju.edu.cn; Tel.: +86-13515817319

**Abstract:** The vertical motion of air is closely related to the amount of precipitation that falls in a particular region. The Tibetan Plateau and the North Pacific are important determinants of the East Asian climate. We use climate diagnosis and statistical analysis to study the vertical motion of the air over the North Pacific and Tibetan Plateau and the relationship between the vertical motion of air over them and the climate in East Asia. Here we show that there is a downward movement of air over the Tibetan Plateau during the winter, with a maximum velocity of downward movement at 500 hPa, whereas there is an upward movement of air with a maximum velocity of upward movement at 600 hPa during the summer. Precipitation in East Asia has a significant negative correlation (The correlation coefficient exceeds −0.463 and confidence level is greater than 99%) with the vertical motion of air over the Tibetan Plateau and the North Pacific during both the winter and summer. There is also a negative correlation of precipitation in the region south of the Yangtze River with the vertical motion of air over the Tibetan Plateau in winter, whereas the area of negative correlation to the vertical motion of air over the North Pacific in winter is located to the east of the Tibetan Plateau and the Yangtze–Huaihe river basin. The research results provide a climatic framework for the vertical motion of air over both the Tibetan Plateau and the North Pacific.

**Keywords:** Tibetan Plateau; North Pacific; vertical motion of air; climate in East Asia

## 1. Introduction

The formation of rainfall is a complex atmospheric process and is influenced by many different factors, one of the most important of which is the upward motion of air.

East Asia, located between the Tibetan Plateau and the Pacific Ocean, is the most populous region in the world. The Tibetan Plateau is the highest plateau on the Earth. Driven by its thermal and dynamic effects, the air over the Tibetan Plateau sinks vertically in winter and rises vertically in summer. The changeover between the ascending and descending motion of air occurs in spring and autumn [1–3]. In addition, the thermal effect of the Tibetan Plateau also affects the surrounding vertical circulation (the Bay of Bengal) [4].

The vertical movement of the atmosphere over the Tibetan Plateau forms many vertical circulations in the nearby areas. It also forms vertical circulations in remote areas, such as the Pacific Ocean, the Indian Ocean, Africa, and the southern hemisphere [3,5–9]. The vertical motion patterns have great interannual variation [9].

The vertical movement of air has an important influence on the climate. The upward movement of the atmosphere corresponds to abundant rain, while the downward

movement corresponds to drought [10,11]. Considering the three cells in the meridional direction, a strongly rising package of air is usually associated with low pressure and abundant precipitation, whereas descending air is related to high pressure and drought conditions [12,13]. From the point of view of the zonal Walker and anti-Walker circulation, abundant rainfall is closely tied to the upward motion of air, whereas drought is related to the downward motion of air over the equatorial Pacific [14]. However, only sporadic studies have shown the influence of vertical motion over the Tibetan Plateau on its surrounding climate [15]. The influence of the vertical circulation of air over the Tibetan Plateau on the climate of the Yangtze River basin in China with a large population has not received enough attention.

The Pacific Ocean is the largest ocean on Earth. As early as the beginning of the last century, Henry [16] indicated that the thermal difference between the Pacific Ocean and the East Asian mainland can affect rainfall in East Asia. Warm winds from the Pacific Ocean may result in increased rainfall, whereas cold winds from the mainland may lead to decreased rainfall. The thermal action from the Pacific Ocean affects not only the vertical circulation in the Pacific, but also the climate in East Asia [17]. The discovery of the El Niño–Southern Oscillation [18,19] showed that sea surface temperature anomalies can cause anomalous convection (vertical velocity) and lead to changes in the climate in East Asia through a teleconnection mechanism [17,20–25], whereas the relationship between the climate of East Asia and the vertical circulation of the atmosphere, which is driven by the thermal action of the Pacific Ocean, receives less attention.

The vertical motion of air over the Tibetan Plateau is driven by thermal and dynamic forces, and the vertical motion of air over the Pacific Ocean is driven by thermal power. Both of these vertical motions affect the climate, but it is not clear what the difference is between the two kinds of effects. Furthermore, it is important to study the influence of the vertical atmospheric movement over the Tibetan Plateau and the Pacific Ocean on the climate of East Asia. The research results are of significance to understand comprehensively the climate of East Asia and make more accurate climate forecasts. The paper is organized as follows. Section 2 describes the data and methods. The results and discussion are presented in Section 3. The study concludes with a brief summary in Section 4.

## 2. Data and Methods

### 2.1. Data

The period measured was 1981–2010. The monthly mean data calculated using these datasets formed the basis of the analytical approach. Data for the monthly mean wind speed were obtained from the National Centers for Environmental Prediction/National Center for Atmospheric Research (NCEP/NCAR) Reanalysis dataset [26] with a resolution of $2.5° \times 2.5°$. Twelve pressure levels were used for the North Pacific (1000, 925, 850, 700, 600, 500, 400, 300, 250, 200, 150 and 100 hPa) and eight levels for the Tibetan Plateau (600–100 hPa). Precipitation data from 839 meteorological stations were provided by the China Meteorological Administration. The surface air temperature and atmospheric air pressure were extracted from the Scientific Data Center for the Cold and Arid Regions of China surface meteorological datasets with a temporal and spatial resolution of $0.1° \times 0.1°$. The location of the Tibetan Plateau was taken as (25–40° N, 75–105° E). Taking into consideration the remote connection between the equatorial Pacific sea surface temperature, the North Pacific climate system and the East Asian climate [17], the location of the North Pacific was taken as (30° S–60° N, 120° E–85° W).

To determine the reliability of the data, we compared the vertical motion of air in the NCEP, ERA-Interim (produced by the European Center for Medium-Range Weather Forecasts), and JRA-55 (from the Japan Meteorological Agency) datasets over the Tibetan Plateau and the Pacific Ocean and found that they have almost identical systems and centers [27]. We also compared the vertical circulation along 90° E, calculated using observational data [6,8] with the NCEP analysis data for plateau regions [9] and found that the vertical motion of air in the Pacific Ocean calculated from the observational data [6–8]

was consistent with that calculated from the NCEP data. It is therefore reasonable to analyze the vertical motion of air using the NCEP data.

*2.2. Methodology*

To diagnose and analyze the vertical motion of air over the Tibetan Plateau and the North Pacific, empirical orthogonal function (EOF) analysis [28,29] was used to decompose the spatial and temporal variation of the original vertical velocity in winter (January) and summer (June). The EOF analysis can be used to decompose the original data field, anomaly field and standardization field of vertical velocity. The results of decomposing different data fields are different in climatic significance. The calculation of the EOF is as follows:

$$X_{m \times n} = V_{m \times p} \times T_{p \times n} \tag{1}$$

where $X_{m \times n}$ is a data matrix of the original vertical velocity composed of *n* observations of m spatial points (if *m* > *n*, transpose *X* before calculating), *p* is the number of spatial eigenvectors. *V* are spatial eigenvectors and *T* are time coefficients.

Here we used the first spatial eigenvector and its corresponding time series in the analysis. To determine whether the first spatial eigenvector has a physical meaning, we used the rule suggested by [29] to test the results.

The time series of the corresponding first spatial eigenvector (spatial mode) in both winter and summer was used to extract the periodic variation signals of the spatial distribution mode using Morlet wavelet analysis [30–32].

The continuous wavelet transform $W_n{}^X(s)$ on a scale s of a discrete time series $x_n$ (n = 1,..., N) with uniform time steps $\delta t$ was defined as the convolution of $x_n$ with the scaled and translated version of the wavelet function $\psi_0$:

$$W_n^X(s) = \sqrt{\frac{\partial t}{s}} \sum_{n'=0}^{N-1} x_{n'} \psi_0^* \left[ \frac{(n' - n)\partial t}{s} \right] \tag{2}$$

where * indicates the complex conjugate, $N$ is the total number of data points in the time series and $(\partial t/s)^{1/2}$ is the factor used to normalize the wavelet function, such that every wavelet function has a unit energy at each wavelet scale *s*.

By transforming the wavelet scale s and localizing along the time index *n*, we obtained a diagram showing the fluctuation characteristics of the time series at a certain scale and its variation with time—that is, the wavelet power spectrum [30,31]. The Morlet wavelet is not only nonorthogonal, but is an exponential complex-valued wavelet regulated by a Gaussian distribution defined as:

$$\psi_0(t) = \pi^{-1/4} e^{i\omega_0 t} e^{-t^2/2} \tag{3}$$

where *t* is the dimensionless time and $\omega_0$ is the dimensionless frequency. When $\omega_0 = 6$ the wavelet scale s is basically equal to the Fourier period ($\lambda = 1.03 s$) [33], so the scale term and the periodic term can be substituted for each other. The wavelet power spectrum $\left| W_n^X(s) \right|^2$ is then calculated [30–32].

To eliminate edge effects (i.e., the cone of influence), we used red noise processes as the background spectrum to test the statistical significance of the wavelet power spectrum [30–32]. Values outside the cone of influence were estimated at the 95% confidence level on each scale.

Correlation analyses were conducted between the time series of the principal mode and the meteorological indices (surface air temperature, atmospheric pressure and precipitation) in January and June; *t*-tests were used to verify the statistical results.

## 3. Results

### 3.1. Vertical Motion of Air over the Tibetan Plateau

3.1.1. Horizontal Distribution of Vertical Velocity over the Tibetan Plateau

To understand fully the distribution of vertical velocity over the Tibetan Plateau, we divided the year into four seasons: spring (March–May); summer (June–August); autumn (September–November); and winter (December–February). We used the data for the seasonal mean vertical velocity over the last 30 years. As the mean altitude of the Tibetan Plateau is >3000 m a.s.l. (700 hPa), we analyzed the seasonal mean vertical velocity at 500 hPa to avoid any effects from the boundary layer (Figure 1a). The two updraft centers in the western (32° N, 82.5° E) and eastern (33° N, 97° E) regions of the Tibetan Plateau clearly show that the upward movement of air dominates the Tibetan Plateau and also affects the surrounding regions during the summer months. The center of the downdrafts during the winter months is located in the southeastern corner (30° N, 100° E) of the Tibetan Plateau.

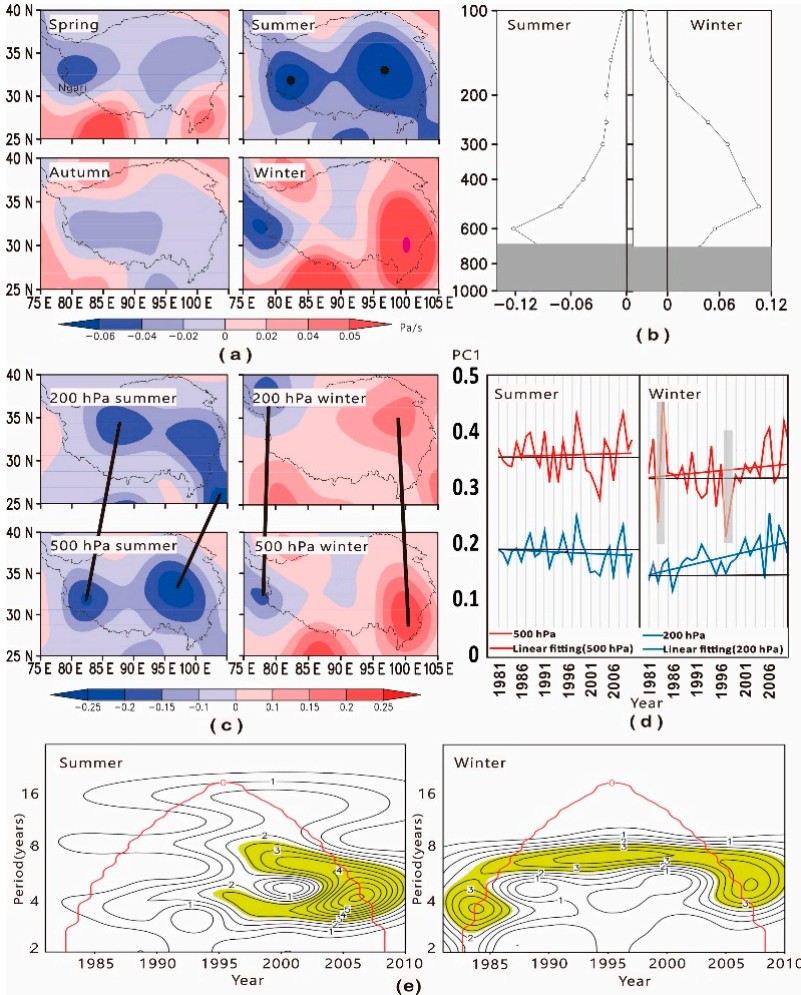

**Figure 1.** Distribution of the vertical velocity of air over the Tibetan Plateau. (**a**) Vertical velocity at 500 hPa in spring (March–May), summer (June–August), autumn (September–November) and winter (December–February). (**b**) Profiles of vertical velocity centers in summer (33° N, 97° E) and winter (100° E, 30° N). (**c**) Spatial distribution of the primary EOF-analyzed mode for vertical velocities at 200 and 500 hPa in summer and winter. (**d**) Temporal variation of the primary mode of the vertical velocities at 200 and 500 hPa in summer and winter. *PC* denotes principal component and gray areas denote large fluctuations. (**e**) Wavelet power spectrum of temporal coefficients of the primary mode of the vertical velocity at 500 hPa in both summer and winter. The red line delineates the cone of influence and the yellow areas show confidence levels >95%.

The movement of air is generally upward over the main body of the plateau during spring and autumn. As can be seen from the 30 years of monthly means of vertical velocity (Figure 2), the downdrafts were dominant in most regions on and surrounding the Tibetan Plateau before March, except in the regions which are near to the Qaidam Basin. In April, the upward motion of air began to dominate the entire Tibetan Plateau. During autumn, the updrafts were overwhelmingly dominant over the Tibetan Plateau before October, but downdrafts suddenly became dominant during November, except for the areas surrounding Ngari Prefecture in Tibet. A sudden transition between the upward and downward movement of air can therefore be observed over the Tibetan Plateau in both April and November.

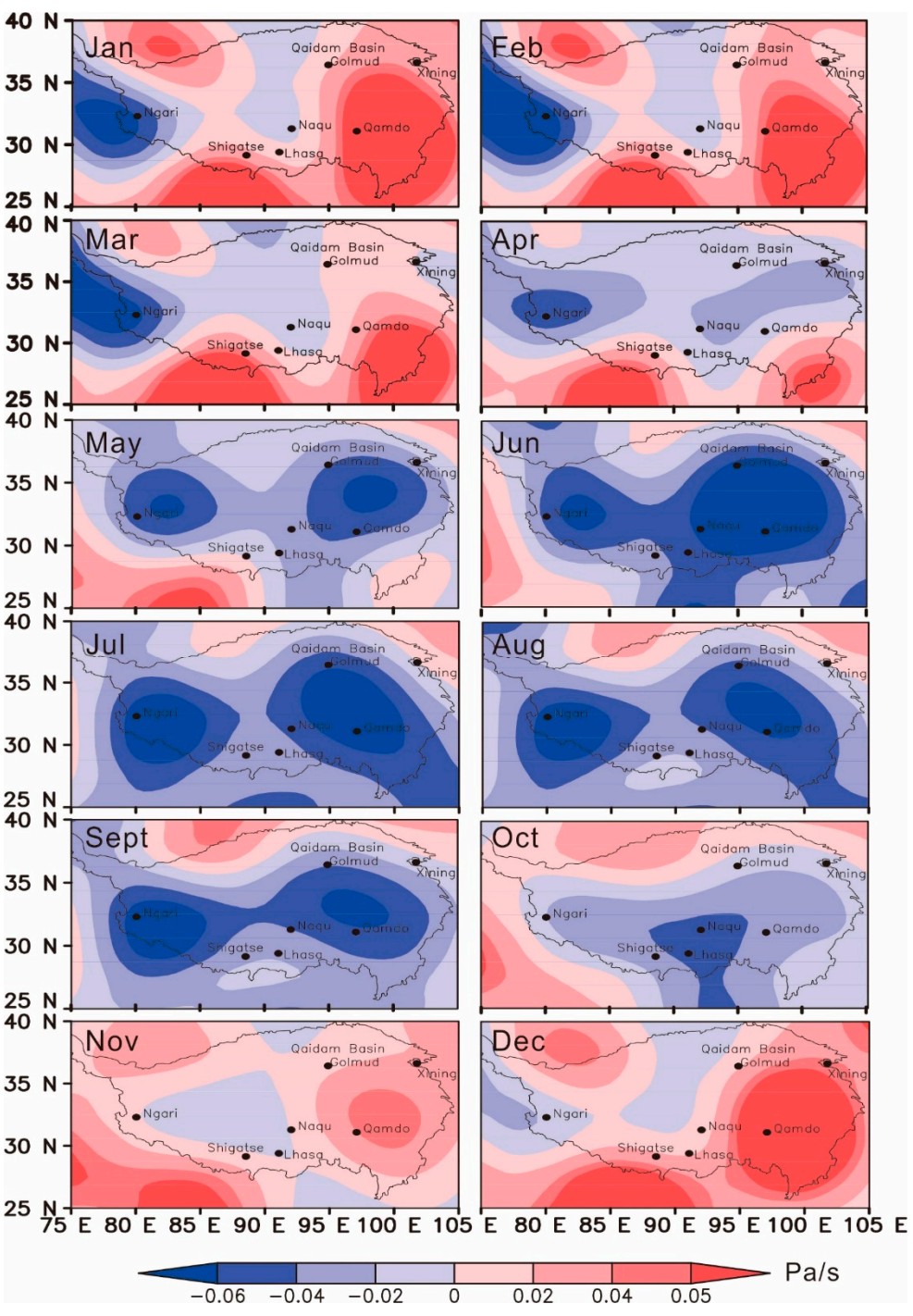

**Figure 2.** Monthly means of vertical velocity over the Tibetan Plateau at 500 hPa from 1981 to 2010.

### 3.1.2. Vertical Distribution of Vertical Velocity over the Tibetan Plateau

Figure 1b shows the vertical distribution of the vertical velocity over the centers of downdraft on the Tibetan Plateau (100° E, 30° N) during winter and the eastern center of updraft (33° N, 97° E) during the summer. Upward movements of air over the eastern center dominated throughout the troposphere during the summer months. The vertical velocity decreased as the altitude increased. The maximum velocity of upward movement was located at 600 hPa and vertical velocity was zero at 100 hPa.

We also studied the South Asian High, which is mainly located around 100–70 hPa with a center at (32–35° N, 60–80° E). The updraft centers (33° N, 97° E) during the summer (Figure 1b) were located at the eastern margin of the South Asian High.

The center of maximum velocity of downward movement was at 500 hPa in winter. Downdrafts dominated below 200 hPa, but updrafts occurred above this level. Downdrafts below 200 hPa indicated that the Tibetan Plateau was controlled by a high-pressure system during the winter, in agreement with the work of Yeh et al. [6].

### 3.1.3. The Principal Modes of Vertical Velocity of Air over the Tibetan Plateau

The maximum downward vertical velocities over the Tibetan Plateau occur at 500 hPa in winter and the height of the transition between ascending and descending motion is near 200 hPa (Figure 1b). Therefore, the vertical velocities at 200 and 500 hPa were used to conduct the EOF analysis to investigate the principal modes of vertical velocity over the Tibetan Plateau and any impact on the surrounding regions.

Since EOF decomposes the original vertical velocity of the air, the spatial distribution of the principal mode represents the average distribution feature of the vertical velocity of air, and its time series represents the time-varying characteristics of the average distribution of the vertical velocity of air. The principal spatial distribution mode shows that the centers of maximum vertical velocity at about 500 and 200 hPa do not coincide in summer and winter. In summer, the 200 hPa center in the west is orientated to the north and east of the 500 hPa center and the 200 hPa center in the east is orientated to the south and east of the 500 hPa center, whereas it is to north and west of the 500 hPa center in winter (Figure 1c). The two vertical axes across the centers in both the east and the west therefore tilt eastward in summer and westward in winter.

Figure 1d shows the time series of the principal model. From the figure, we know that these time coefficients are all greater than zero. This may be because we decomposed the original vertical velocity (X matrix in Equation (1)) by EOF analysis, the time series of the principal mode came to the first quadrant after EOF decomposition (coordinate rotation) [34–38]. The same applies to the EOF analysis for the vertical velocity over the Pacific Ocean. The variations of the principal mode for the vertical motion of air in the plateau region show that the vertical velocity increased in winter in the time period 1981–2010 at both 200 and 500 hPa (Figure 1d). We defined the change value of time series (PC1) by subtracting the previous value from the latter value. When the change value of the PC1 (absolute value) exceeded the average value over 10%, we defined the fluctuation to be large. The large fluctuations in the time series were in 1983 and 1997 (Figure 1d) which both correspond to El Niño events (Table 1).

### 3.1.4. Period of Variation in the Vertical Velocity over the Tibetan Plateau

The explained variances in the vertical velocity for the primary mode analyzed using the EOF method at the 500 hPa level were 92% in summer and 88% in winter and therefore their time series can be used to reflect the variation characteristics of the original vertical velocity field over the Tibetan Plateau in these two seasons. To further understand the variation period in the vertical velocity field, Morlet wavelet analysis was conducted to extract the relevant time coefficients. Morlet wavelet analysis rendered a primary modal period of 3–4 years for summer from 1995 to 2010 and for winter before 1987, and a secondary period of 6–8 years for winter from the mid-1980s to 2010 (both significant at the 95% confidence level) (Figure 1e).

**Table 1.** The El Niño events and the large fluctuations of time series of vertical motion at over the North Pacific.

| Time of El Niño Events | Maximum Warming Zone | Time of Large Fluctuations | | | |
|---|---|---|---|---|---|
| | | 500 hPa in Winter | 850 hPa in Winter | 500 hPa in Summer | 850 hPa in Summer |
| | | 1981–1982 | 1981–1982 | | |
| 1982.05–1983.08 | NINO3 | | | | |
| 1986.09–1988.02 | NINO3,4 | | | | |
| | | 1988–1989 | 1988–1989 | | |
| 1990.08–1991.03 | NINO4 | 1990–1991 | | | |
| 1991.05–1992.05 | NINO3,4 | | | | |
| | | | 1992–1993 | | |
| 1994.06–1995.04 | NINO4 | | | | |
| | | | 1996–1997 | | |
| 1997.05–1998.05 | NINO3 | 1997–1998 | 1997–1998 | | |
| | | 1998–1999 | 1998–1999 | 1998–1999 | |
| | | 1999–2000 | 1999–2000 | | |
| | | 2000–2001 | 2000–2001 | | |
| 2002.01–2003.04 | NINO4 | 2002–2003 | | | |
| 2003.07–2004.02 | NINO4 | | | | |
| 2004.06–2005.05 | NINO4 | | | | |
| | | | 2005–2006 | | |
| 2006.08–2007.02 | NINO4 | | | 2006–2007 | |
| 2009.06–2010.04 | NINO4 | | | | |

Note: El niño data from [39].

The periodic phenomena of 3–4 years and 6–8 years may be caused by the periodic change of sensible heat and latent heat on the Tibetan plateau [40–42]. More details need to be studied further. This difference between summer and winter may be attributed to different mechanisms leading to the vertical motion of air. Further investigations indicated that the vertical motion of air during the winter is mainly dynamically induced, whereas vertical motion of air in summer is primarily thermally forced [1,5,43].

### 3.2. Vertical Movement of Air over the North Pacific

We analyzed the vertical motion of air over the North Pacific (30° S–60° N, 120° E–85° W) (Figure 3). There is a marked meridional circulation during winter (Figure 3a), characterized by updrafts at about 45° N and downdrafts at about 30° N, updrafts in the tropics (from the equator to 10–15° N), downdrafts in the mid-latitudes (about 30° N) (the Hadley Cell) [44]. This is consistent with Huang and Yeh [6,25]. However, this meridional circulation only appears in the central and eastern Pacific Ocean during spring, summer and autumn. Updrafts occur in the western Pacific Ocean and downdrafts occur in the eastern Pacific Ocean in these three seasons, suggesting the existence of a zonal circulation (Figure 3a). There is a center of upward movement in the western Pacific Ocean (north of the equator and southeast of Guam, over the southwestern Marshall Islands) in all four seasons and a center of downward movement in the eastern (to the west of California) North Pacific.

We considered the impact of the sea surface temperature in the tropical Pacific on the circulation system in the high-latitude Pacific and the climate in East Asia [17]. We selected the entire North Pacific for EOF analysis.

We select 850 hPa and 500 hPa for analysis, because the subtropical high (caused by the sinking motion of the Hadley Cell and Ferrel Cell) in the North Pacific Ocean has an important influence on the climate of East Asia. Furthermore, the most obvious subtropical high is at 500 hPa, and the most stable subtropical high is at 850 hPa [13,45]. Figure 3b shows the results of the EOF analyses of the original vertical velocity at 500 and 850 hPa for both winter and summer.

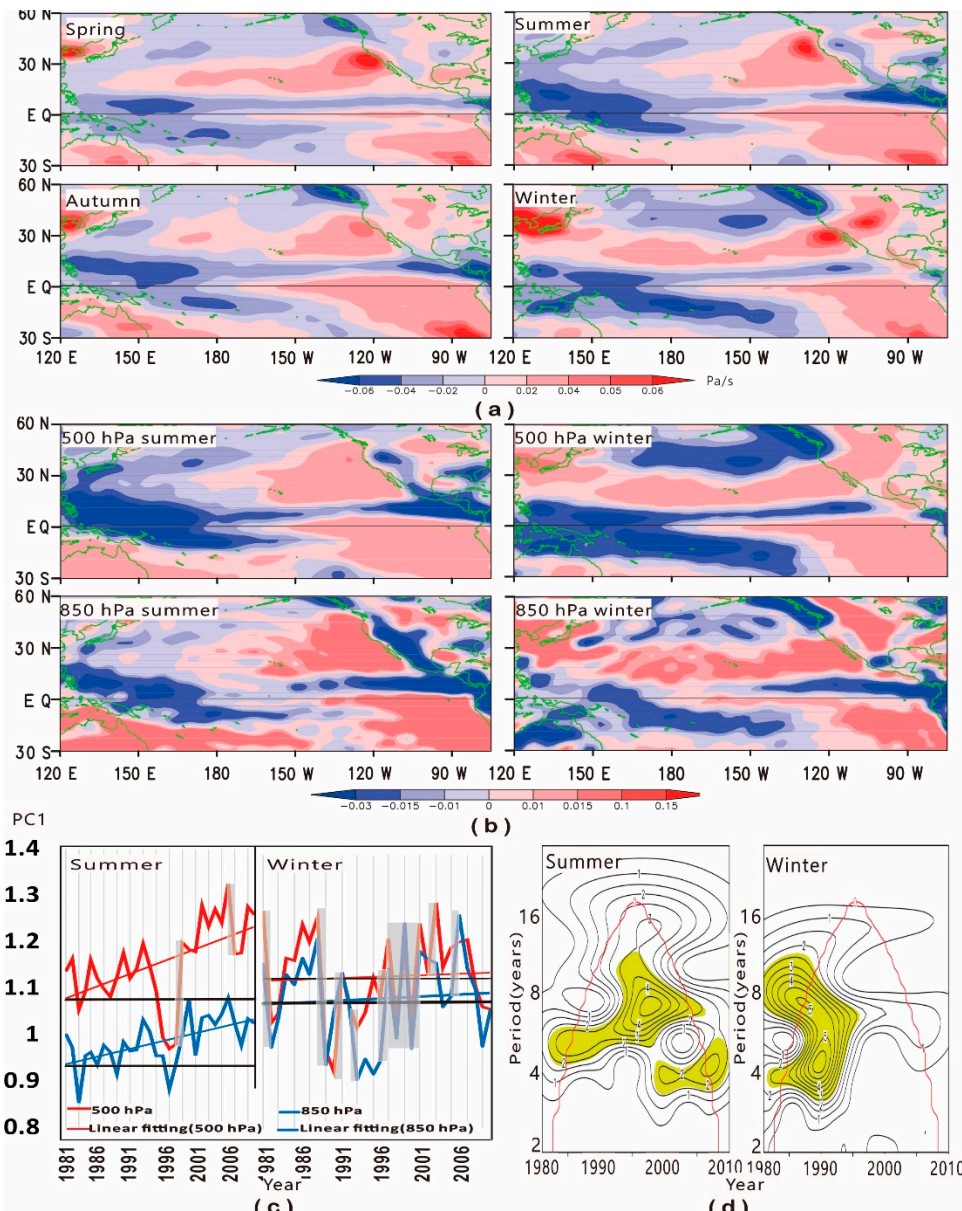

**Figure 3.** Distribution of the vertical velocity of air over the North Pacific. (**a**) Vertical velocity at 500 hPa in spring (March–May), summer (June–August), autumn (September–November) and winter (December–February). (**b**) Primary EOF-analyzed mode of vertical velocities at 500 and 850 hPa in summer and winter. (**c**) Temporal variation in the primary mode of the vertical velocities at 500 and 850 hPa in summer and winter. PC denotes principal component and gray areas denote large fluctuations. (**d**) Wavelet power spectrum of the temporal coefficients of the primary mode of vertical velocity at 500 hPa in both summer and winter. The red line delineates the cone of influence and the yellow areas show confidence levels >95%.

A meridional circulation can be identified in the eastern Pacific Ocean in summer, although upward movement is dominant in the west. This may constitute a zonal circulation.

The time series for the primary mode (with explained variations of 78 and 83%, respectively) shows a particularly noticeable upward trend during summer (Figure 3c). In other words, the upward movement in the west and the downward movement in the east strengthened in summer from 1981 to 2010. The large fluctuation of the time series occurred within the two years before and after an El Niño event (Table 1).

The wavelet analysis for the time series of the primary mode at the 500 hPa level shows that the vertical velocity over the Pacific Ocean has a 4–5-year cycle in winter before

1995, and 4–5-year cycles after 2000 and 7–8-year cycles before 2005 in summer (Figure 3d). The period of 4–5 years may be caused by the periodic change of sea surface temperature in the equatorial eastern Pacific [46–51], and the 7–8-year cycle may be caused by the subtropical ocean circulations in the North Pacific subtropical gyre [52–54].

### 3.3. Vertical Motion of Air over the Tibetan Plateau and the East Asian Climate

To investigate the influence of the vertical motion of air over the Tibetan Plateau on the East Asian climate, we conducted correlation analyses between the time series of the primary EOF-analyzed 500 hPa vertical velocity mode and local Chinese meteorological variables (Figure 4a). The meteorological variables included the surface air temperature, surface air pressure and precipitation. As the minimum monthly temperature in the Tibetan Plateau region occurred in January, the maximum in June and July, and the largest amount of warming in June [55], the meteorological variables in January and June were therefore selected for further study.

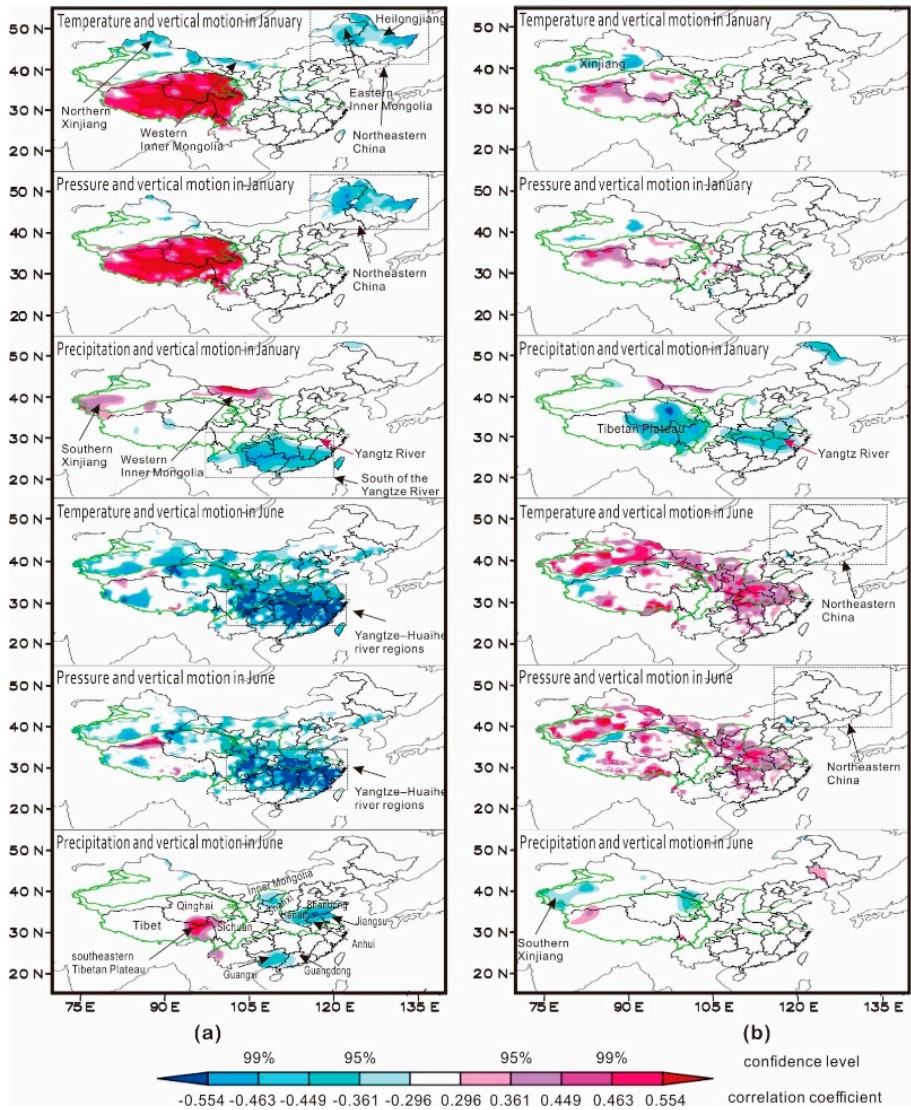

**Figure 4.** Correlation analysis between the surface meteorological variables and the vertical motion of air for (**a**) the Tibetan Plateau and (**b**) the North Pacific. Red shades denote a positive correlation and blue shades denote a negative correlation. Note: the South China Sea has not been marked due to layout reasons.

### 3.3.1. Vertical Motion of Air over the Tibetan Plateau in January and its Relation with the East Asian Climate

- Vertical velocity and climate on the Tibetan Plateau.

Figure 4a shows a positive correlation between the surface air temperature and the time series of the primary mode for vertical velocity at 500 hPa over the Tibetan Plateau in January. The *t*-test correlation coefficients for most regions exceed the critical 95% confidence level and some even exceed the 99% confidence level. The positive correlations indicate that the surface temperature increased as the time series increased (vertical motions strengthened). Because most of the Tibetan Plateau was controlled by downdrafts during January, the positive correlations indicate that when the downdrafts of air strengthened, the surface temperature increased, and vice versa.

- Vertical velocity and climate in the regions surrounding the Tibetan Plateau.

The correlations between the time series of the primary mode of vertical velocity at 500 hPa over the Tibetan Plateau in January and the meteorological variables in the regions surrounding the Tibetan Plateau are described in the following sections.

- Surface temperatures and vertical motion of air.

The time series of primary mode for downdrafts over the Tibetan Plateau in January are negatively correlated with the surface temperature in both northeastern (i.e., the provinces of Heilongjiang and eastern Inner Mongolia) and northwestern (i.e., western Inner Mongolia and northern Xinjiang Province) China from 1981 to 2010 (Figure 4a). Sinking is dominant over the Tibetan Plateau in January, the time series of the principal mode (about 77% of the total variance) increases, the corresponding subsidence motion also increases. The negative correlation shows that when the subsidence motion strengthens over the Tibetan Plateau, the surface air temperature in northeastern China shows the reverse change—that is, the surface air temperature decreases, and vice versa.

- Surface pressure and vertical motion of air.

The correlations between the vertical motion of air over the Tibetan Plateau and the surface pressure over the major parts of China's mainland in January is similar to temperature—namely, there is a negative correlation between the downdrafts over the Tibetan Plateau and the surface pressure in northeastern and northwestern China (Figure 4a). This indicates that the pressure in these regions shows the opposite trend to the time coefficient of the first EOF—that is, the surface pressure in these parts of China decreases as the downdrafts over the Tibetan Plateau increase (Figure 4a). An analysis of the synoptic fields [56] shows that a trough appears over northeastern China, whereas there is a ridge over the Tibetan Plateau. Updrafts over northeastern China increase as the vertical downdrafts over the Tibetan Plateau strengthen. We suggest that there is a good degree of teleconnection.

- Precipitation and vertical motion of air.

The vertical motion of air over the Tibetan Plateau shows a significant correlation with precipitation in some other regions of China in January. There is a significant negative correlation between precipitation in the regions south of the Yangtze River and the vertical motion of air over the Tibetan Plateau in January, with reduced levels of precipitation in the regions south of the Yangtze River corresponding to strengthening downdrafts over the Tibetan Plateau. Precipitation in the southern Xinjiang and western Inner Mongolia shows a positive correlation with the vertical motion of air over the Tibetan Plateau in January, indicating that precipitation in these regions increases with the enhanced downdrafts over the Tibetan Plateau in January (Figure 4a).

### 3.3.2. Vertical Motion of Air over the Tibetan Plateau in June and its Relationship with the East Asian Climate

● Surface temperature and the vertical motion of air.

The surface temperature in most regions of East Asia shows a negative correlation with the vertical motion of air over the Tibetan Plateau in June, especially in the Yangtze–Huaihe river regions. The *t*-test correlation coefficients exceed the critical 99% confidence level, indicating that the surface temperatures in these areas decrease as the updrafts over the Tibetan Plateau increase (Figure 4a).

● Surface pressure and the vertical motion of air.

The correlation between the vertical motion of air over the Tibetan Plateau and the surface pressure in most regions of East Asia is similar to that for surface temperatures. The surface pressure in the major parts of China's mainland decreases as updrafts over the Tibetan Plateau in June strengthen, especially in the Yangtze–Huaihe river regions. The *t*-test correlation coefficients exceed the critical 99% confidence level (Figure 4a).

● Precipitation and the vertical motion of air.

Four significant precipitation regions are found in China in June. There is a region of positive correlation between precipitation over the southeastern Tibetan Plateau (on the borders of Tibet, Qinghai and Sichuan) and the vertical motion of air over the Tibetan Plateau in June (Figure 4a), suggesting that precipitation in these regions increases as the updrafts over the Tibetan Plateau increase. Three regions with a negative correlation between precipitation in the most of China's mainland and the vertical motion of air over the Tibetan Plateau in June were located in the border region between northern Shanxi Province and Inner Mongolia, the eastern provinces of Shandong, Henan, Anhui and Jiangsu in China, and the southern Chinese provinces of Guangdong and Guangxi. These three regions of descending air compensate for the ascent of air over the Tibetan Plateau during June [9], suggesting that strengthened updrafts over the Tibetan Plateau contribute toward reduced amounts of precipitation in these three regions.

### 3.4. Vertical Motion of Air over the North Pacific and the East Asian Climate

To investigate the relationship between the East Asian climate and the vertical motion of air over the North Pacific, correlation analyses were conducted between the time series coefficients of the primary EOF-analyzed mode for vertical velocity at 500 hPa and the surface temperature, surface pressure and precipitation in the major parts of China's mainland (Figure 4b).

### 3.4.1. Vertical Velocities over the North Pacific in January and the Climate in East Asia

● Surface temperature (pressure) and the vertical motion of air over the North Pacific.

The time series of the primary mode for vertical motion of air over the North Pacific in January shows a positive correlation with the surface temperature (pressure) on the Tibetan Plateau and a negative correlation in Xinjiang, China (Figure 4b). This suggests that the surface temperature (pressure) on the Tibetan Plateau shows the same change trend as the vertical motion of air over the North Pacific, but the opposite change trend to Xinjiang. This means that any increase in the vertical motion of air at 500 hPa over the North Pacific corresponds to an increase in surface temperature (pressure) on the Tibetan Plateau and a reduced surface temperature (pressure) in Xinjiang.

● Precipitation and the vertical motion of air.

The time series of the vertical motion of air over the North Pacific in January shows a significant negative correlation with precipitation in the middle to lower reaches of the Yangtze River and on the eastern Tibetan Plateau (Figure 4b), indicating that precipitation decreases as the vertical motion of air at 500 hPa over the North Pacific strengthens.

### 3.4.2. Vertical Velocities of Air over the North Pacific and the June Climate in East Asia

● Surface temperature and pressure and the vertical motion of air over the North Pacific.

The vertical motion of air over the North Pacific in June shows a positive correlation with both surface temperature and pressure for most regions in East Asia (Figure 4b). Specifically, any increase in the vertical motion of air at 500 hPa over the North Pacific corresponds to an enhanced surface temperature and pressure in most regions of China. The subtropical high over the North Pacific is mainly the result of the combined downdraft of the Hadley Cell and part of the Ferrel Cell. Every summer, the subtropical high of the North Pacific will stretch westward and uplifts as a result of the influence of thermal difference between land and ocean, affecting the climate of China [57]. When the subtropical high of the North Pacific strengthens westward and extends northward it will cause an increase in pressure and temperature in these area.

● Precipitation and the vertical motion of air over the North Pacific.

In contrast with January, the vertical motion of air over the North Pacific in June shows a significantly negative correlation with precipitation in a few regions, including the northeastern Tibetan Plateau and southern Xinjiang in China (Figure 4b). Precipitation in these regions decreases as the vertical motion of air at 500 hPa over the North Pacific strengthens.

### 3.5. Difference of the East Asian Climate to Vertical Motion of Air over the Tibetan Plateau and North Pacific

### 3.5.1. Surface Temperature and Pressure

The relationship of surface temperature and pressure in the different regions of East Asia to the vertical motion of air over the Tibetan Plateau and the North Pacific display different characteristics in summer and winter (Figure 4a,b). There is a significant difference between northeastern China and the Tibetan Plateau in January. The surface temperature and pressure in northeastern China are negatively correlated with the vertical motion of air over the Tibetan Plateau (i.e., the surface temperature and pressure in northeastern China decrease as the vertical motion of air over the Tibetan Plateau increases), whereas the vertical motion of air over the North Pacific is not significantly correlated with that in northeastern China.

The surface temperature and pressure on the Tibetan Plateau are positively correlated with the vertical motion of air over the Tibetan Plateau and North Pacific in January. The correlation between the surface temperature (pressure) on the Tibetan Plateau and the vertical motion of air over the Tibetan Plateau exceeds that observed for the North Pacific. The *t*-test correlation coefficients for the former exceeded the critical 99% confidence level for most regions, whereas the *t*-test correlation coefficients for the latter exceeded the critical 95% confidence level for only some regions (Figure 4a,b).

The vertical motion of air over the Tibetan Plateau and the North Pacific in June show opposing correlations with the surface temperature (pressure) in most of China's mainland. The vertical motion of air over the Tibetan Plateau displays a negative correlation with the surface temperature (pressure) in the most of China's mainland, but the correlation between the vertical motion of air over the North Pacific and the surface temperature (pressure) is positive in June (Figure 4).

### 3.5.2. Precipitation

Precipitation in East Asia in January shows different correlations to the vertical motion of air over the Tibetan Plateau and the North Pacific. Precipitation in the regions south of the Yangtze River are negatively correlated with the vertical motion of air over the Tibetan Plateau—that is, precipitation decreases as downdrafts strengthen over the Tibetan Plateau (Figure 4a). By contrast, precipitation in the middle and lower reaches of the Yangtze River and on the eastern Tibetan Plateau are negatively correlated with the vertical motion of

air over the North Pacific—that is, precipitation decreases as the vertical motion of air strengthens (Figure 4b).

The correlations of precipitation in East Asia to the vertical motion of air over the Tibetan Plateau and the North Pacific in June is not as significant as that in January and displays only sporadic correlation. Precipitation in the Yellow River region and south of the Yangtze River in China shows a negative correlation with updrafts over the Tibetan Plateau—that is, precipitation decreases as the strength of the updrafts over the Tibetan Plateau increases. Precipitation on the southeastern Tibetan Plateau in China shows a positive correlation with the updrafts, increasing as the strength of any upward movement over the Tibetan Plateau increases (Figure 4a). The vertical motion of air over the North Pacific shows a negative correlation with precipitation in southern Xinjiang and the northeastern Tibetan Plateau, whereas it shows a positive correlation with precipitation in the west of the plateau (Figure 4b).

## 4. Discussions

### 4.1. Positive Correlation between the Vertical Motion of Air and the Surface Temperature and Pressure over the Tibetan Plateau in January.

The correlation coefficients between the vertical motion of air and the surface temperature (pressure) over the Tibetan Plateau exceed the critical 95% confidence level (*t*-test) for most of the regions of the plateau. There are two possible mechanisms:

(1) When the subsidence motion of air increases in January, the cloud amount decreases, the direct solar radiation increases, and the net radiation at the ground surface increases. Therefore, the radiation energy used to heat the atmosphere increases by sensible heat [57], which increases the surface temperature, and forms a positive correlation between the vertical motion of air and the surface temperature over the Tibetan Plateau.

(2) Sinking air from high latitudes is dominant over the Tibetan Plateau in January (Figure 5a). The atmosphere over the plateau sinks in January. According to the first law of thermodynamics in P coordinate (Equation (4)) [58], the main factors affecting the temperature change in a certain place are temperature advection, vertical motion and non-adiabatic heating. It is known from the influence term of vertical motion $(\gamma_d - \gamma)\omega$ that when the atmospheric stratification is stable $(\gamma_d - \gamma > 0)$, the sinking motion $(\omega > 0)$ will cause local warming. It is a dry season in the Tibetan Plateau during winter, and the air stratification is relatively stable. When there is a sinking movement of air, it will cause local warming. At the same time, as the downdrafts strengthen, the mass of air per unit area increases, leading to increases in surface pressure:

$$\frac{\partial T}{\partial t}_p = -u\frac{\partial T}{\partial x}_p - v\frac{\partial T}{\partial y}_p + \frac{\omega}{\rho g}(\gamma_d - \gamma) + \frac{\varepsilon}{c_p\rho} \tag{4}$$

where, $\varepsilon = \rho\frac{\delta q}{\delta t}$, δq: external heating item to unit mass of air, $\rho$: the density of air.

### 4.2. Correlation of Surface Temperature (Pressure) to the Vertical Motion of Air in June

The surface temperature (pressure) in different regions of most of China's mainland in June shows a reverse correlation with the vertical motion of air over both the Tibetan Plateau and the North Pacific. A significant negative correlation between surface temperature (pressure) and the vertical motion of air over the Tibetan Plateau appears in the Yangtze and Yellow River regions of eastern China (the *t*-test correlation coefficients exceed the critical 99% confidence level) (Figure 4a).

The low pressure vortex over the Tibetan Plateau often occurs in summer, most of which moves eastward out of the plateau, affecting the weather in eastern China [59]. The low pressure vortex over the plateau strengthens and moves eastward, inducing a decrease in pressure over eastern China. This low pressure from the Tibetan Plateau increases cloud amount and precipitation in the eastern China region, which further contributes to reduced surface temperatures in those regions.

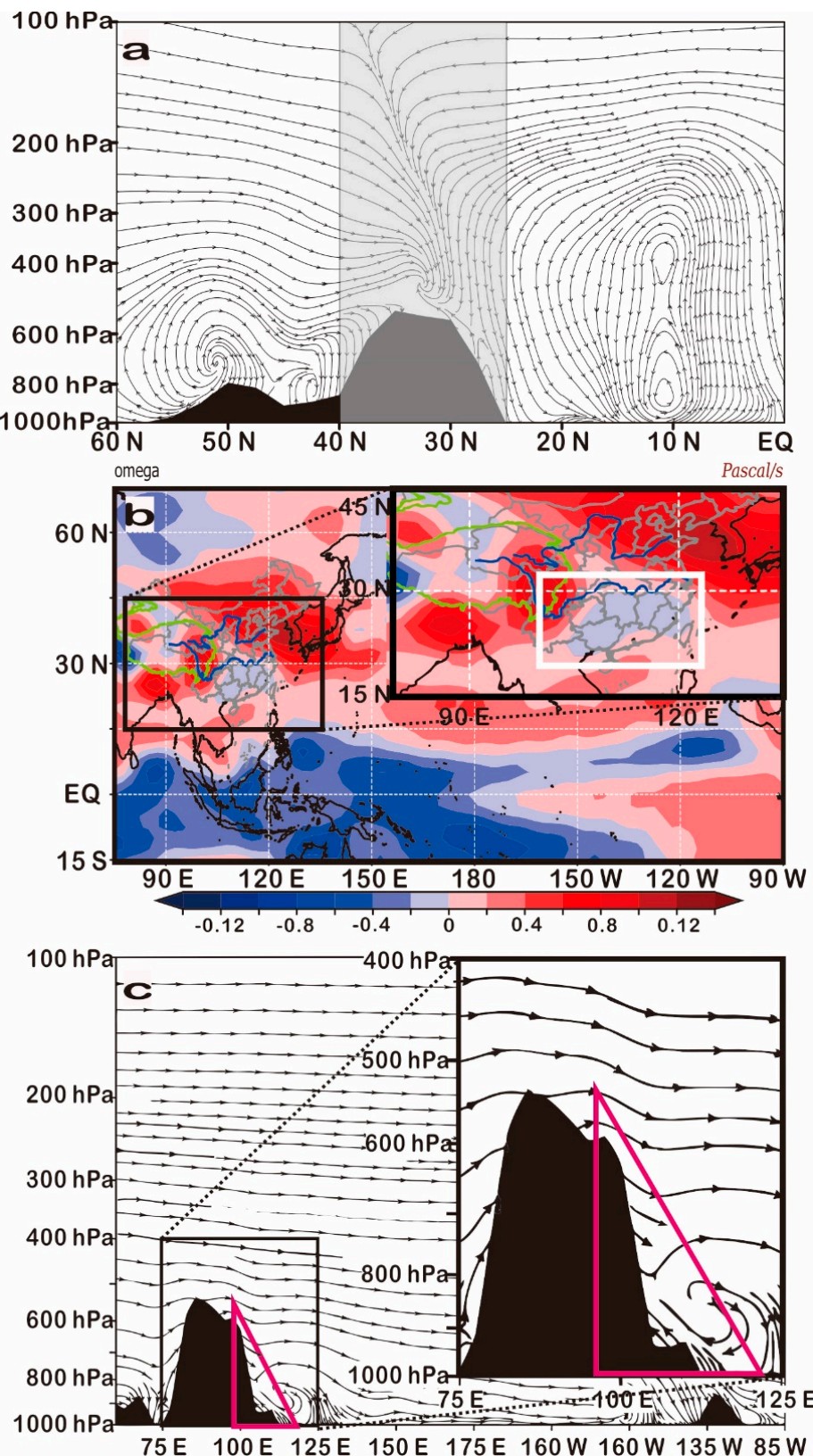

**Figure 5.** Wind velocity fields. (**a**) Mean meridional circulation at 90° E in January. (**b**) Vertical velocity field at 500 hPa in January. (**c**) Mean zonal circulation at 30° N in January. The grey area in part (**a**) represents the Tibetan Plateau area, and the white rectangle (**b**) represents the downdraft area and the red triangle (**c**) represents the intersection of the updraft and the downdraft.

There is a positive correlation between the vertical motion of air over the North Pacific and the surface temperature and pressure in eastern China in June (Figure 4b). The main weather system affecting the climate of East Asia in summer is the Pacific subtropical high (the downdraft from the Hadley and Ferrel circulations) [58]. The subtropical high's extension westward and jump northward will affect the weather in China. Warm downdrafts dominate when this subtropical high controls the Yangtze River region, causing the surface temperature and pressure to increase.

*4.3. Correlations of Precipitation to the Vertical Motion of Air over Tibetan Plateau in January*

The negative correlation between precipitation in East Asia and the vertical motion of air over the Tibetan Plateau in January occurs in the regions south of the Yangtze River (south of 30° N) (Figure 4a), which is consistent with the vertical upward region of air at 500 hPa (Figure 5b). Figure 5c shows that the sinking of air dominates in the upper atmosphere of these negative correlation regions, and the ascending airflow is dominant in the lower atmosphere in the regions south of the Yangtze River, and these sinking airflows from the Tibetan Plateau (Figure 5c). The downdrafts in the upper atmosphere of these negative correlation regions intensify as the sinking over the Tibetan Plateau enhances, causing precipitation in the Yangtze–Huaihe river regions to decrease, and vice versa.

When the subtropical high stretches westward and uplifts northward, the downdraft it brings reduces the precipitation in the middle and lower reaches of the Yangtze River, forming a negative correlation.

*4.4. Explanation on Correlation Coefficients between Precipitation and Vertical Motion over the Tibetan Plateau and North Pacific*

From the previous analysis, we know that although precipitation in East Asia has a significant negative correlation with the vertical motion of air over the Tibetan Plateau and the North Pacific during both the winter and summer, most of the correlation coefficients are between 0.449 and 0.554. The possible reasons are as follows:

The correlation coefficients in this study are restricted to linear correlations [60]. Whereas, the relationship between precipitation and vertical air motion over the Tibetan Plateau and the Pacific Ocean may be nonlinear. For example, precipitation in the south of the Yangtze River Basin in China is not only correlated with the vertical motion of air over the Tibetan Plateau, but also with the vertical motion of air over the Pacific Ocean (with a lower correlation coefficient (Figure 4b)), and may be affected by other factors that we do not know. The vertical movement of the air over the Tibetan Plateau may also be influenced by the Pacific Ocean, and the vertical movement of the air over the Pacific Ocean may also be affected by the Tibetan Plateau, and so on. The impact factors are variable and the relationship is complicated. The scope of this study cannot cover all of this details and I would like to suggest further research in this topic. Our study gives the correlation coefficient >0.463 when the confidence level is 99%. The correlation coefficient is reasonable.

This study only shows the correlation between the vertical movement of air and China's climate, what causes the vertical movement is not known. The mechanism behind this phenomenon will need further study. In addition, we can only show cross-sections along 90°E and 30°N (Figure 5a,c), due to the limitation of our drawing level. The above results would be more clearly shown if a flow profile could be drawn from the origin area of high latitude cold air to the Tibetan Plateau and then from the plateau to the negatively correlated area south of the Yangtze River. There are some phenomena we cannot explain yet. These deficiencies will need to be addressed in future research.

**5. Conclusions**

The study examined the distribution and variation of the vertical motion of air over the Tibetan Plateau and the North Pacific Ocean, as well as the correlation between their variation and China's climate. This study will provide us with the distribution characteristics and variation rules of the vertical motion of air over the Tibetan Plateau and the

North Pacific Ocean, and the relationship and differences between the climate of China and the vertical motion of air over the Tibetan Plateau and the North Pacific Ocean. The results of this study will provide background knowledge and reference for China's climate prediction. The details are as follows:

(1) Updrafts dominate the Tibetan Plateau in the boreal spring, summer and autumn, ascending to the maximum value of $-0.1245$ Pa/s at 600 hPa in the boreal summer. By contrast, downdrafts dominate in winter, descending to a maximum value of 0.1087 Pa/s at 500 hPa. The vertical motion of air over the North Pacific has generally strengthened over the past 30 years.

(2) The surface temperature, pressure, and precipitation on most of China's mainland all have significant correlations (the correlation coefficient exceeds $-0.463$ and confidence level is greater than 99%) to the vertical motion of air over the Tibetan Plateau and the North Pacific. This study compared the differences in correlation of the Tibetan Plateau and the North Pacific. The surface temperature and pressure in June (summer) generally have a negative correlation to the vertical motion of air over the Tibetan Plateau, that is, surface temperature and pressure decrease with the strengthening of the vertical movement of air, whereas a positive correlation over the North Pacific, namely, surface temperature and pressure increase with vertical motion, and vice versa.

(3) The correlation of precipitation to the vertical motion of air in January (winter) changes with the geophysical location. Precipitation in regions south of the Yangtze River have a negative correlation to the vertical motion of air over the Tibetan Plateau. That is, precipitation decreases with the increase of the vertical movement. The eastern Tibetan Plateau and the Yangtze–Huaihe river regions have a negative correlation to the vertical motion of air over the North Pacific. Precipitation decreases when the downdrafts of air strengthen, and vice versa.

**Author Contributions:** R.T. conceived the idea and wrote the manuscript. X.Z. and D.Z. performed the processing data and statistical analysis. Y.M. and W.M. revised the manuscript. All authors have read and agreed to the published version of the manuscript.

**Funding:** This research was funded by the Second Tibetan Plateau Scientific Expedition and Research Program (STEP), Grant No. 2019QZKK0103 and the National Natural Science Foundation of China (Grant Nos 41775142).

**Institutional Review Board Statement:** Not applicable.

**Informed Consent Statement:** Not applicable.

**Data Availability Statement:** Data and methods used in the research have been presented in sufficient detail in the paper.

**Acknowledgments:** We acknowledge the use of meteorological data collected from the National Center for Environmental Prediction, the China Meteorological Administration and the Scientific Data Center for the Cold and Arid Regions of China. All data in the research can be obtained by contacting the corresponding author, Rongxiang Tian (trx@zju.edu.cn).

**Conflicts of Interest:** The authors declare no conflict of interest.

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
