# Peer review of "Longer Time-Scale Variability of Atmospheric Vertical Motion over the Tibetan Plateau and North Pacific and the Climate in East Asia"

_atmosphere, doi:10.3390/atmos12050630_

Round 1

Reviewer 1 Report

The author revised the manuscript properly. So It is acceptable for the publication.

Author Response

Thank you a lot for your comments!

Reviewer 2 Report

Thanks to the Authors for their answers

Author Response

Thank you a lot for your comments! 

Reviewer 3 Report

This text is much improved from the first revision, although it still requires a lot of general editing. There is much information here, and it is easy to get lost in the detail: the authors need to make the text more reader friendly by providing clearer guidance, more general information and tidying up the MS. My greatest concern is the interpretation of the statistical results. Although there is high confidence that the results are correct, the correlation co-efficients cited for the various processes and relationships are not that high. It is essential that this is clearly flagged in the text and, ideally, discussion of any alternative mechanism  presented (or it is shown clearly how and why such mechanisms do not need to be considered).

Line 17: delete ‘the’.

Line 19: delete comma.

Line 22: you should give the correlation value and the significance level.

Line 25: change to ..’winter, whereas the..’.

Line 27: don’t use ’great significance’ unless you are quoting a statistical value.

Lines 27/28: change to ‘to understand comprehensively’ to remove split infinitive.

Line 28: insert ‘a’ after ‘make’.

Line 35: capital ‘E’ required for ‘earth’.

Lien 37: ‘conversion of’  is the wrong words here, I think you mean ‘changeover between’.

Lines 41/43 (and elsewhere) you need a topographic map to show the places referred to in the text. The non-Chinese reader will have no idea where many of the places mentioned are (e.g. line 167).

Line 44: don’t use ‘etc’, give all the details.

Line 45: ‘atmospheres’ in this context does not make sense.

Line 52: where is the upward motion?

Line 54: change to ‘studies have shown the..’.

Line 58: capital ‘E’ required for ‘earth’.

Line 66: change to ‘], whereas the relationship..’

Line 67: insert ‘is’ after ‘which’.

Line 68: insert comma after ‘Ocean’.

Lines 69/71: text does not make (English) sense.

Line 74: change to ‘to understand comprehensively’ to remove split infinitive.

Line 75: change ‘forecast’ to ‘forecasts’.

Line 76: delete ‘our’.

Lines 92: move sentence to line 80, before ‘Data’.

Line 93: replace ‘our’ with ‘the’.

Line 97 (and elsewhere): you have used British English elsewhere ibn the MS, change ‘center(s)’ to ‘centre(s) for consistency.

Line 98: insert comma after ‘E’.

Lines 99/100: delete ‘they were consistent. The’ and replace with ‘the’.

Lines 107/108

: cumbersome text, replace with ‘winter (January) and summer (June).’

Line 114: replace ‘If’ with ‘if’.

Line 124: replace full stop with colon.

Lines 137/138: move ‘then’ to after ‘is’ and replace ‘the’ with ‘the’. You need a reference to the calculation of the wave power spectrum (it is not clear from the text how this was done).

Line 182: have you evidence from synoptic charts/observations that there was high pressure at (all) those times?

Line 184: the units are missing from the scale bar of Figure 1c, and you need ‘years’ on the horizontal axes of Figures 1d and e. The vertical axis of Figure 1d needs a label.

Line 202: change ‘velocity’ to ‘velocities.

Line 221: delete ‘time’.

Line 222: how were ‘large’ fluctuations defined, and what was the rationale for choosing these criteria? Change ‘is’ to ‘are’.

Line 227: capital ‘N’ required for ‘niño’.

Line 240: you need to give more details about this, just providing a reference is not adequate (the reader should not have to hunt through the reference for this sort of detail). I am unclear if this should be references [39, 40, 41] or just [39, 41].

Line 250: Capital ‘C’ for ‘cell’. Start the sentence with ‘This’ and delete ‘result’.

Line 258: insert ‘the’ after ‘in’.

Line 258/265: this text needs editing for (English) sense. I can’t suggest any changes as I can change the meaning of the text in several ways and I don’t know what you are trying to say.

Lines 267/268: delete ‘The observed…Pacific Ocean’ and replace with ‘This’.

Line 274: change to ‘Niño’.

Line 281: see comments regarding missing units and labels in Figure 1.

Line 293: insert space between number and unit.

Line 302: you ned to move Figure 4 closer to where it is referred to in the text. Currently it is five pages away.

Line 304: insert space between number and unit.

Line 315: ‘Surface…air’ is not a sentence. Make it a clearer sub-heading (it will also help guide the reader through a lot of information in the following pages).

Line 325: see comment for Line 315.

Line 336: see comment for Line 315.

Line 345: I have a concern with the correlations in Figure 4. The correlation co-efficients range from -0.5 to +0.5. Whilst the confidence levels may be quite high, the corelations are not that strong (medium). On this basis here and elsewhere) you are attributing a lot of processes to cause and effect. Some indication of this (and consideration of possible other influences/factors) should be mentioned. If the correlations were >=0.7 I would be less concerned.

Line 349: see comment for Line 315 (and there is a font style change).

Line 355: see comment for Line 315 (and there is a font style change).

Line 361: see comment for Line 315 (and there is a font style change). Also, insert ‘precipitation’ after ‘significant’.

Lines 365/373: there is far too much detail here! The reader won’t have any idea where these areas are without a map (see previous comments).

Line 377: delete ‘the’ after ‘Pacific’.

Line 383: see comment for Line 315 (and there is a font style change).

Line 392: see comment for Line 315 (and there is a font style change).

Line 399: see comment for Line 315 (and there is a font style change).

Line 404: insert ‘the’ after first ‘of’ and ‘Cell’ after ‘Hadley’. Capital C; for ‘cell’ after ‘Ferrel’.

Line 408: delete comma after ‘northward’.

Line 417: see comment for Line 315 (and there is a font style change).

Line 343: define ‘the major parts’.

Line 436: define ‘the major parts’.

Line 349: see comment for Line 315 (and there is a font style change).

Lines 460/461: see comment for Line 315 (and there is a font style change).

Line 463: which regions did not show the 99% confidence level?

Line 472: I am  ot clear what the P co=ordinate is, and the value P (written as such) does not appear in equation 4 (although the text implies that it does).

Line 476: replace ‘its’ with ‘the’.

Line 482: see comment for Line 315 (and there is a font style change).

Line 483: again, you need to make it clear which the major parts are, and why they are considered major.

Line 496: cumbersome English. I would say ‘The low pressure vortex over the Tibetan..’.

Lines 497/499: poorly written. Not sure what the meaning of ‘updrafts’ in brackets is. ‘moves’ should be ‘and moved’. Should it be ‘over; rather than ‘in the eastern China region (and why not just ‘over eastern China’?).

Line 503: again, not clear about the text in brackets.

Line 505: insert ‘the’ before ‘Hadley’, change ‘Ferrer; to ‘Ferrel’ and change ‘circulation’ to ‘circulations’.

Line 514: change ‘parts’ to ‘part’.

Line 517: see comment for Line 315 (and there is a font style change).

Line 521: insert space between number and unit.

Line 526: insert ‘and’ before ‘vice’.

Line 530/531: text does not make sense as written. I would delete the ‘and’ in line 531 and add ‘is not known’ after ‘movement. Delete the question mark.

Line 551: give the maximum value (and ideally the range). Information in the conclusions should stand alone.

Line 554: see above, quantify ‘significant’.

Line 560, change ‘..motion. And..’ to ‘..motion, and’.

Line 563: put a full stop after ‘Plateau’ and start a new sentence.

Line 566: put a full stop after ‘Pacific’ Delete ‘namely‘ , start a new sentence with ‘Precipitation’.

Line 567: insert ‘and’ before ‘vice’.

Author Response

Thank you a lot for your comments!My reply is attached。

Round 2

Reviewer 3 Report

In my comments on the previous draft of this MS I expressed two main concerns. First, the text required a substantial review and revision for English and clarity. Second, I noted that the authors had presented or interpreted their statistical results in a misleading way.

The first of these issues has been addressed (largely), although there are several errors which have slipped through, and revised/added text has introduced new errors. These are noted below.

However, I still have concerns about the statistical presentation of the results, which are misleading. As I noted previously the correlation coefficients are not strong. From what I can tell they lie mostly between about +/-0.4 and +/-0.55. Correlation coefficients between +/-0.5 and +/-0.7 are moderately correlated, correlations below +/-0.5 are poorly correlated (see e.g. https://www.andrews.edu/~calkins/math/edrm611/edrm05.htm). This needs to come out much more clearly in the presentation of the results. At present the text is presented along the lines of ‘there is a positive or negative correlation between X and Y’, with no discussion of the strength of this and what might be causing it (and, as a general comment, Figure 4 where most of these results are presented is not easy to interpret the way the data are set out). The high confidence levels of the correlations has no bearing on the strength of the correlations. The fact that these correlations are weak must be acknowledged and discussed in the text.  I flagged this in my previous comments. The authors responded that this was an issue for further work, but the tone of the text is largely unchanged. Unless these issues are addressed I cannot recommend publication.

Line 25: quantification of correlation required (this is the Abstract)

Line 49: change ‘strong’ to ‘strongly’

Line 68: change ‘get’ to ‘receives’

Line 81: delete ‘time’ to remove tautology (time is a period)

Line 98: change ‘centers’ to ‘centres’ ( this escaped the previous edits of this error)

Lines 162/163: why is the text in italics, and lower case M required for ‘monthly’

Line 164: change ‘is’ to ‘are’

Line 183: space required between number and unit

Line 192: which is the text in italics

Line 197: line required between number and unit

Line 201: change ‘occurs’ to ‘occur’

Line 203: change ‘velocity’ to ‘velocities’

Lines 223/225: why is the text in italics?

Line 224: delete ‘Then’ and start sentence with ‘When’

Line 225: delete ‘is’

Line 230: space required between number and unit in column headers

Lines 244/245: why is the text in italics?

Line 260: delete ‘the’ in red text

Line 263: insert ‘the’ after ‘in’

Line 264: change ‘Asian’ to ‘Asia’

Line 266:  spaces required between number and unit

Line 448: delete ‘the ‘ after ‘in’

Line 495: replace full stop with colon

Line 504: lower case T for ‘the’

Line 506: insert ‘the’ after ‘over’

Lines 519 et seq.:  spaces required between numbers and units on vertical axes of the diagrams

Line 561:  space required between number and unit

Line 565: delete ‘the’ before ‘most’

Author Response

Comments and Suggestions for Authors

In my comments on the previous draft of this MS I expressed two main concerns. First, the text required a substantial review and revision for English and clarity. Second, I noted that the authors had presented or interpreted their statistical results in a misleading way.The first of these issues has been addressed (largely), although there are several errors which have slipped through, and revised/added text has introduced new errors. These are noted below.However, I still have concerns about the statistical presentation of the results, which are misleading. As I noted previously the correlation coefficients are not strong. From what I can tell they lie mostly between about +/-0.4 and +/-0.55. Correlation coefficients between +/-0.5 and +/-0.7 are moderately correlated, correlations below +/-0.5 are poorly correlated (see e.g. https://www.andrews.edu/~calkins/math/edrm611/edrm05.htm). This needs to come out much more clearly in the presentation of the results. At present the text is presented along the lines of ‘there is a positive or negative correlation between X and Y’, with no discussion of the strength of this and what might be causing it (and, as a general comment, Figure 4 where most of these results are presented is not easy to interpret the way the data are set out).The high confidence levels of the correlations has no bearing on the strength of the correlations.

The fact that these correlations are weak must be acknowledged and discussed in the text.  

I flagged this in my previous comments. The authors responded that this was an issue for further work, but the tone of the text is largely unchanged. Unless these issues are addressed I cannot recommend publication.

Response: Thank you very much for your comment and advice.

(1) The correlation coefficient is not strong.

Whether the correlation coefficient is significance must pass the reliability test (Mudelsee, 2010; Filliben, 1975). Table 1 shows the critical correlation coefficients under different confidence levels. As shown in Table 1, when the confidence level is 99% and the degree of freedom is 50, the critical value of the correlation coefficient is 0.354 (rc =0.354). When the degree of freedom is 10, rc =0.708; when the degree of freedom is 5, the critical value of the correlation coefficient is rc =0.875. That is, under the condition of 99% confidence level, only when the correlation coefficient is larger than or equal to the critical value can it be statistically significant. That is to say, under 99% reliability factor, when (i) the degree of freedom is 50 and the correlation coefficient is larger than or equal to 0.354 (r≥ rc), (ii) the degree of freedom is 10 and the correlation coefficient is larger than or equal to 0 .708, and (iii) the degree of freedom is 5 and the correlation coefficient is larger than or equal to 0.875, we know that these three cases (i) (ii) (iii) have the same statistical significance (Mudelsee, 2010; Filliben, 1975). The time series length of our manuscripts was 30 years, and the correlation coefficient (r>0.449) was statistically significant.

(2) The relationship between precipitation and vertical air movement over the Tibetan Plateau and the Pacific Ocean may be nonlinear, and only their linear relationships have been studied in this manuscript.

(3) The vertical movement of the air over the Tibetan Plateau is also influenced by the Pacific Ocean, and vice versa. The relationship between them will be remained to be further studied.

Table 1 Critical correlation coefficient under different confidence levels

degrees of freedom

confidence level

95%

99%

1

0.997

1.00

2

0.95

0.99

3

0.878

0.959

4

0.811

0.917

5

0.755

0.875

6

0.707

0.834

7

0.666

0.798

8

0.632

0.765

9

0.602

0.735

10

0.576

0.708

11

0.553

0.684

12

0.532

0.661

13

0.514

0.641

14

0.497

0.623

15

0.482

0.606

16

0.468

0.59

17

0.456

0.575

18

0.444

0.561

19

0.433

0.549

20

0.423

0.537

21

0.413

0.526

22

0.404

0.515

23

0.396

0.505

24

0.388

0.496

25

0.381

0.487

26

0.374

0.479

27

0.367

0.471

28

0.361

0.463

29

0.355

0.456

30

0.349

0.449

31

0.344

0.442

32

0.339

0.436

33

0.334

0.430

34

0.329

0.424

35

0.325

0.418

36

0.32

0.413

37

0.316

0.408

38

0.312

0.403

39

0.308

0.398

40

0.304

0.393

50

0.273

0.354

100

0.195

0.254

Notes: this table is made according to student's t test (Mudelsee, 2010; Filliben, 1975)

References:

Mudelsee, Manfred. Climate time series analysis: [M]. Springer Netherlands, 2010, 156-319

Filliben, J. The probability plot correlation coefficient test for normality. Technometrics, 1975, 17, 111–117.

Line 25: quantification of correlation required (this is the Abstract)

Response: Thank you very much for your comment and advice.

They have been rewritten in line 23.

Line 49: change ‘strong’ to ‘strongly’

Response: Thank you very much for your comment and advice.

 ‘strong’ has been replaced by ‘strongly’

Line 68: change ‘get’ to ‘receives’

Response: Thank you very much for your comment and advice.

‘get’ has been replaced by ‘receives’

Line 81: delete ‘time’ to remove tautology (time is a period)

Response: Thank you very much for your comment and advice.

 ‘time’ has been deleted.

Line 98: change ‘centers’ to ‘centres’ ( this escaped the previous edits of this error)

Response: Thank you very much for your comment and advice.

 ‘centers’ has been changed to ‘centres’.

Lines 162/163: why is the text in italics, and lower case M required for ‘monthly’

Response: Thank you very much for your comment and advice.

They have been corrected.

Line 164: change ‘is’ to ‘are’

Response: Thank you very much for your comment and advice.

 ‘is’ has been replaced by ‘are’.

Line 183: space required between number and unit

Response: Thank you very much for your comment and advice.

The space has been added between number and unit.

Line 192: which is the text in italics

Response: Thank you very much for your comment and advice.

It's marked in red and italics

Line 197: line required between number and unit

Response: Thank you very much for your comment and advice.

The space has been added between number and unit.

Line 201: change ‘occurs’ to ‘occur’

Response: Thank you very much for your comment and advice.

 ‘occurs’ has been replaced by ‘occur’

Line 203: change ‘velocity’ to ‘velocities’

Response: Thank you very much for your comment and advice.

 ‘velocity’ has been replaced by ‘velocities’.

Lines 223/225: why is the text in italics?

Response: Thank you very much for your comment and advice.

They have been corrected.

Line 224: delete ‘Then’ and start sentence with ‘When’

Response: Thank you very much for your comment and advice.

 ‘Then, when’ has been replaced by ‘When’.

Line 225: delete ‘is’

Response: Thank you very much for your comment and advice.

 ‘is’ has been deleted.

Line 230: space required between number and unit in column headers

Response: Thank you very much for your comment and advice.

The space has been added between number and unit.

Lines 244/245: why is the text in italics?

Response: Thank you very much for your comment and advice.

They have been corrected.

Line 260: delete ‘the’ in red text

Response: Thank you very much for your comment and advice.

 ‘the’ has been deleted.

Line 263: insert ‘the’ after ‘in’

Response: Thank you very much for your comment and advice.

 ‘the’ has been inserted after ‘in’.

Line 264: change ‘Asian’ to ‘Asia’

Response: Thank you very much for your comment and advice.

 ‘Asian’ has been replaced by ‘Asia’.

Line 266:  spaces required between number and unit

Response: Thank you very much for your comment and advice.

The space has been added between number and unit.

Line 448: delete ‘the ‘ after ‘in’

Response: Thank you very much for your comment and advice.

 ‘the’ after ‘in’ has been deleted.

Line 495: replace full stop with colon

Response: Thank you very much for your comment and advice.

The full stop has been replaced by colon.

Line 504: lower case T for ‘the’

Response: Thank you very much for your comment and advice.

 ‘The’ has been replaced by ‘the’.

Line 506: insert ‘the’ after ‘over’

Response: Thank you very much for your comment and advice.

 ‘the’ has been inserted after ‘over’.

Lines 519 et seq.:  spaces required between numbers and units on vertical axes of the diagrams

Response: Thank you very much for your comment and advice.

The spaces have been added between numbers and units on vertical axes of the diagrams.

Line 561:  space required between number and unit

Response: Thank you very much for your comment and advice.

The space has been added between number and unit.

Line 565: delete ‘the’ before ‘most’

Response: Thank you very much for your comment and advice.

 ‘the’ has been deleted.

This manuscript is a resubmission of an earlier submission. The following is a list of the peer review reports and author responses from that submission.

Round 1

Reviewer 1 Report

This study investigated the relationship between vertical motion over the Tibetan Plateau and North Pacific and East Asia climate. It is somewhat interesting, but there are some major concerns related to the physical mechanism for the relationship. The concerns are as follows; 

1. The authors explained the positive correlation between the vertical motion of air and the surface temperature and pressure over the Tibetan Plateau in January by using the thermodynamic equation. But I cannot understand the authors' explanation because the sinking motion at a pressure level can lead to local warming at the level rather than the surface. That is, adiabatic warming by sinking motion could not be the reason for the surface warming over the Tibetan Plateau in January. The authors had better explain the physical mechanism with radiation effect rather than adiabatic warming. In a number of studies, the pressure system over the Tibetan Plateau has been explained based on the heat flux, which can be associated with snow depth. Therefore, the authors should consider such possible mechanisms for the relationship.       

2. For the correlation of surface pressure and temperature to the vertical motion over the Tibetan Plateau, the authors mentioned that the low pressure transports cold air from the Tibetan Plateau, further contributing to reduced surface temperature in the eastern China region. Is this right? Then the authors should add some references for this or show the additional analysis. Or the surface temperature in summer can be related to the cloud amount or precipitation. Thus, cold temperature can be induced by increasing cloud amount or precipitation, which resulted from the low-pressure system moving eastward.   

3. For the correlation of precipitation in East Asia to the vertical motion of air over the Tibetan Plateau in January, what is the possible physical mechanism?

4. Why the authors analyzed only for 30 years (1981-2010)? I just wonder that this study's results (e.g., correlations) are still the same if the analysis period is extended to 40 years (1981-2020). I have checked that the monthly NNRP1 data is available for 40 years on the website (https://psl.noaa.gov/data/gridded).

Author Response

Please see the attachment。

Reviewer 2 Report

This paper is particularly interesting for an audience in the field of atmosphere physics and climatology  but may be as well interesting for the broader audience of atmospheric sciences

At research, the purpose is difficult to identify.

The research area, adopted in the publication, requires depiction on pictorial drawing, including locations of measuring station covered in the research, especially the Fig1 (line 232), since there is already an outline of this area. In the text (Page 4 line 158) other regions next to the Tibetan Plateau was appeared, which should be indicated in the figure.

Homogeneity of data is crucial issue in analisys but is not confirmed in any way.

Do the meteorological data included in the study also come from aerological measurements?

Specific Comments for Authors

 Page 2 line 84

Are the included data (e.g. precipitation data) homogeneous? In what way was their homogeneity verified? - I suggest adding the information at the research

Page 2 line 92

The research period, i.e. 1981-2010 - why data from recent years were not taken into account?

Page 3 line 94

„To determine the reliability of the data, we compared the vertical motion…” – I suggest completing the significance level at which the data reliability was determined

Page 3 line 100, 101

page 4 line 163

I suggest adding these drawings in your research;

Page 4 line 167

Page 10 line 315,316,339

Page 11 line 360, 365,366 i inne

The places and regions mentioned in the text, should be indicated on the drawing presenting the research area

Page 16 Fig4b

The given scale means ‘correlation coefficient’?

Reviewer 3 Report

Detailed comments have been provide to the Editors, for sharing with the authors if appropriate.
